# Enhancing Neural Network Performance with Leader-Follower Architecture and Local Error Signals

## Abstract

The collective behavior of a network with heterogeneous, resource-limited information processing units (e.g., group of fish, flock of birds, or network of neurons) demonstrates high self-organization and complexity. These emergent properties arise from simple interaction rules where certain individuals can exhibit leadership-like behavior and influence the collective activity of the group. Driven by the natural collective ensembles, we introduce a *worker* concept to artificial neural network (NN). This NN structure contains workers that encompass one or more information processing units (e.g., neurons, filters, layers, or blocks of layers). Workers are either leaders or followers, and we train a leader-follower neural network (LFNN) by leveraging local error signals. LFNN does not require backprobagation (BP) and global loss to achieve the best performance (we denote LFNN trained without BP as LFNN-$\ell$). We investigate worker behavior and evaluate LFNN and LFNN-$\ell$ through extensive experimentation. On small datasets such as MNIST and CIFAR-10, LFNN-$\ell$, trained with local error signals achieves lower error rates than previous BP-free algorithms and even surpasses BP-enabled baselines. On ImageNet, LFNN-$\ell$ demonstrates superior scalability. It achieves higher accuracy than previous BP-free algorithms by a significant margin. Furthermore, LFNN-$\ell$ can be conveniently embedded in classic convolutional NNs such as VGG and ResNet architectures. Our experimental results show that LFNN-$\ell$ achieves at most 2x speedup compared to BP, and significantly outperforms models trained with end-to-end BP and other state-of-the-art BP-free methods in terms of accuracy on CIFAR-10, Tiny-ImageNet, and ImageNet.

## 1 Introduction

Artificial neural networks (ANNs) typically employ global error signals for learning Rumelhart et al. (1985). While ANNs draw inspiration from biological neural networks (BNNs), they are not exact replicas of their biological counterparts. ANNs consist of artificial neurons organized in a structured layered architecture Thomas & McClelland (2008). Learning in such architectures commonly involves gradient descent algorithms Bottou et al. (1991) combined with backpropagation (BP) Rojas (1996). Conversely, BNNs exhibit more intricate self-organizing connections, relying on specific local connectivity Markram et al. (2011) to enable emergent learning and generalization capabilities even with limited and noisy input data. Simplistically, we can conceptualize a group of neurons as a collection of *workers* wherein each worker receives partial information and generates an output, transmitting it to others so as to achieve a specific collective objective. This behavior can be observed in various biological systems, such as decision-making among a group of individuals Moscovici & Zavalloni (1969), flocking behavior in birds to avoid predators and maintain flock health O'Loan & Evans (1999), or collective behavior in cells fighting infections or sustaining biological functions Friedl et al. (2004).

The study of collective behavior in networks of heterogeneous agents, ranging from neurons and cells to animals, has been a subject of research for several decades. In physical systems, interactions among numerous particles give rise to emergent and collective phenomena, such as stable magnetic orientations Hopfield (1982). A system of highly interconnected McCulloch-Pitts neurons McCulloch & Pitts (1943) has collective computational properties Hopfield (1982). Networks of neurons with graded response (or sigmoid input-output relation) exhibit collective computational properties similar to those of networks with two-state neurons Hopfield (1984). Recent studies focus on exploring

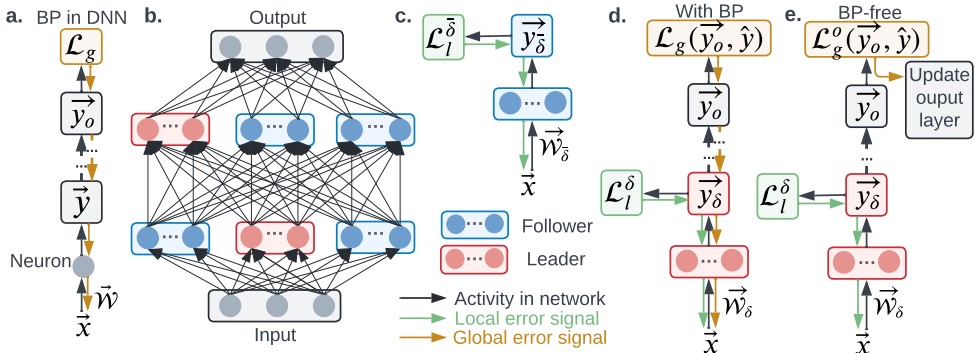

Figure 1: **Weight updates of LFNN. a.** BP in classic deep neural network (DNN) training. Global prediction loss is back-propagated through layers. **b.** An LF hierarchy in a DNN. Within a layer, neurons are grouped as (leader and follower) workers. **c.** Weight update of follower workers. **d.** Weight update of leader workers with BP. **e.** BP-free weight update of leader workers.

collective behaviors in biological networks. This includes the examination of large sensory neuronal networks Tkačik et al. (2014), the analysis of large-scale small-world neuronal networks Qu & Wang (2017), the investigation of heterogeneous NNs Luccioli & Politi (2010), and the study of hippocampal networks Meshulam et al. (2017). These studies aim to uncover the collective dynamics and computational abilities exhibited by such biological networks.

In biological networks such as the human brain, synaptic weight updates can occur through local learning, independent of the activities of neurons in other brain regions Caporale & Dan (2008); Yin et al. (2023). Partly for this reason, local learning has been identified as effective means to reduce memory usage during training and to facilitate parallelism in deep learning architectures, thereby enabling faster training Xiong et al. (2020).

Drawing inspiration from collective behavior and local learning in biological networks, we propose a neural network (NN) architecture mirroring the complexity observed in biological systems. Our *leader-follower neural network* (LFNN) divides the NN into layers of elementary *leader* and *follower* workers, leveraging characteristics of collective motion to designate leadership. Similar to a flock of birds, leaders in LFNNs are informed by and guide the entire system's motion. This biologically-plausible alternative to backpropagation (BP) utilizes distinct error signals for leaders and followers, enabling training through local error signals.

We evaluate LFNN and its BP-free version trained with local loss (LFNN-$\ell$) on MNIST, CIFAR-10, and ImageNet datasets. Our LFNN and LFNN-$\ell$ outperform other biologically plausible BP-free algorithms and achieved comparable results to BP-enabled baselines. Notably, our algorithm demonstrates superior performance on ImageNet compared to all other BP-free baselines. Moreover, LFNN-$\ell$ can be conveniently incorporated into VGG and ResNet to accelerate the training process, and it significantly outperforms state-of-the-art block-wise learning BP-free methods on CIFAR-10, Tiny-ImageNet, and ImageNet. This study, introduces complex collectives to deep learning, provides valuable insights into biologically plausible NN research, and opens up avenues for future work.

## 2 LFNNs Inspired by Complex Collectives

Collective motion refers to ordered movement in systems consisting of self-propelled particles, such as flocking or swarming behavior Bouffanais (2016). The main feature of such behavior is that an individual particle is dominated by the influence of others and thus behaves entirely differently from how it might behave on its own Vicsek & Zafeiris (2012). A classic collective motion model, the Vicsek model Vicsek et al. (1995), describes the trajectory of an individual using its velocity and location, and uses stochastic differential/ difference equations to update this agent's location and velocity as a function of its interaction strength with its neighbors. Inspired by collective motion, we explore whether these minimal mathematical relations can be exploited in deep learning.

**LF hierarchy in fully connected layers.** In a fully-connected (FC) layer containing multiple neurons, we define *workers* as structures containing one or more neurons grouped together. Unlike classic NNs where neurons are the basic units, LFNN workers serve as basic units. By adapting the Vicsek model terms to deep learning, a worker's behavior is dominated by that of neighbors in the same layer. In addition, we consider *leadership* relations inside the group. According to collective motion,

"leadership" involves "the initiation of new directions of locomotion by one or more individuals, which are then readily followed by other group members" Krause et al. (2000). Thus, in FC layers, one or more workers are selected as leaders, and the rest are "followers" as shown in Figure 1b.

**LF hierarchy extended in convolutional layers.** Given a convolutional layer with multiple filters (or kernels), workers can be defined as one or more filters grouped together to form *filter-wise workers*. For a more coarsely-grained formulation, given a NN with multiple convolutional layers, a set of convolutional layers can be grouped naturally as a block (such as in VGG Simonyan & Zisserman (2014), ResNet He et al. (2016), Inception Szegedy et al. (2015) architectures). Our definition of the worker can be easily adapted to encompass *block-wise workers* to reflect this architecture where a block of convolutional layers work together as a single, block-wise worker. Similarly, if a block contains one layer, it becomes a *layer-wise worker*.

More formally, we consider a NN with $\mathcal{M}$ hidden layers, and a hidden layer contains $\mathcal{N}$ workers. A worker can contain one or more individual working components, which can be neurons, filters in convolutional layers, or blocks of NN layers, and each individual working component is parametrized by a set of trainable parameters $\mathcal{W}$. During training, at each time step $t$, leader workers $\mathcal{N}_\delta$ are dynamically selected and the remaining workers are labeled as followers (denoted as $\mathcal{N}_{\bar{\delta}}$) at time step $t$. Following the same notation, leader and follower workers are parameterized by matrices $\vec{\mathcal{W}}_\delta$ and $\vec{\mathcal{W}}_{\bar{\delta}}$, respectively. The output of leader and follower workers in a hidden layer reads $f(\vec{x}, [\vec{\mathcal{W}}_\delta, \vec{\mathcal{W}}_{\bar{\delta}}])$, where $\vec{x}$ is the input to the current hidden layer and $f(\cdot)$ is a mapping function.

**Error signals in LFNN.** In human groups, one key difference between leaders and followers is that leaders are *informed* individuals that can guide the whole group, while followers are uninformed and their instructions differ from treatment to treatment Faria et al. (2010). Adapting this concept to deep learning, LFNN leaders are informed that they receive error signals generated from the global or local prediction loss functions, whereas followers do not have this information. Specifically, assume that we train an LFNN with BP and a global prediction loss function $\mathcal{L}_g$. Only leaders $\mathcal{N}_\delta$ and output neurons receive information on gradients as error signals to update weights. This is similar to classic NN training, so we denote these pieces of information as *global error signals*. In addition, a local prediction error $\mathcal{L}_l^\delta$ is optionally provided to leaders to encourage them to make meaningful predictions independently.

By contrast to leaders, followers $\mathcal{N}_{\bar{\delta}}$ do not receive error signals generated in BP. Instead, they align with their neighboring leaders. Inspired by collective biological systems, we propose an "alignment" algorithm for followers and demonstrate its application in an FC layer as follows: Consider an FC layer where the input to a worker is represented by $\vec{x}$, and the worker is parameterized by $\vec{\mathcal{W}}$ (i.e., the parameters of all neurons in this worker). The output of a worker is given by $\vec{y} = f(\vec{\mathcal{W}} \cdot \vec{x})$. In this context, we denote the outputs of a leader and a follower as $\vec{y}_\delta$ and $\vec{y}_{\bar{\delta}}$, respectively. To bring the followers closer to the leaders, a local error signal is applied between leader workers and follower workers, denoted as $\mathcal{L}_l^{\bar{\delta}} = \mathcal{D}(\vec{y}_\delta, \vec{y}_{\bar{\delta}})$, where $\mathcal{D}(a, b)$[1] measures the distance between $a$ and $b$. In summary, the loss function of our LFNN is defined as follows:

$$\mathcal{L} = \mathcal{L}_g + \lambda_1 \mathcal{L}_l^\delta + \lambda_2 \mathcal{L}_l^{\bar{\delta}} \tag{1}$$

where the first term of the loss function applies to the output neurons and leader workers. The second and third terms apply to the leader and follower workers, as illustrated in Figure 1c and d. The hyper-parameters $\lambda_1$ and $\lambda_2$ are used to balance the contributions of the global and local loss components. It is important to note that the local loss $\mathcal{L}_l^\delta$ and $\mathcal{L}_l^{\bar{\delta}}$ are specific to each layer, filter, or block and do not propagate gradients through all hidden layers.

**BP-free version (LFNN-$\ell$).** To address the limitations of BP such as backward locking, we propose a BP-free version of LFNN. The approach is as follows: In Eq. 1, it can be observed that the weight updates for followers are already local and do not propagate through layers. Based on this observation, we modify LFNN to train in a BP-free manner by removing the BP for global prediction loss. Instead, we calculate leader-specific local prediction loss ($\mathcal{L}_l^\delta$) for all leaders. This modification means that the global prediction loss calculated at the output layer, denoted as $\mathcal{L}_g^o$ (where $o$ stands for output), is only used to update the weights of the output layer. In other words, this prediction loss serves as a local loss for the weight update of the output layer only. The total loss function of the BP-free

---

[1]In our experimentation, we utilize mean squared error loss.

LFNN-$\ell$ is given as follows:

$$\mathcal{L} = \mathcal{L}_g^o + \mathcal{L}_l^\delta + \lambda \mathcal{L}_l^{\bar{\delta}} \tag{2}$$

By eliminating the backpropagation of the global prediction loss to hidden layers, the weight update of leader workers in LFNN is solely driven by the local prediction loss, as depicted in Figure 1e. It's important to note that the weight update of follower workers remains unchanged, regardless of whether backpropagation is employed or not, as shown in Figure 1c.

*Dynamic leadership selection.* In our LF hierarchy, the selection of leadership is dynamic and occurs in each training epoch based on the local prediction loss. In a layer with $\mathcal{N}$ workers, each worker can contain one or more neurons, enabling it to handle binary or multi-class classification or regression problems on a case-by-case basis. This unique characteristic allows a worker, even if it is located in hidden layers, to make predictions $\vec{y}$. This represents a significant design distinction between our LFNN and a traditional NN. Consequently, all workers in a hidden layer receive their respective prediction error signal, denoted as $\mathcal{L}_l^\delta(\vec{y}, \hat{y})$. Here, $\mathcal{L}_l(\cdot, \cdot)$ represents the prediction error function, the superscript $\delta$ indicates that it is calculated over the leaders, $\hat{y}$ denotes the true label, and the top $\delta$ ($0 \leq \delta \leq 100\%$) workers with the lowest prediction error are selected as leaders.

**Definition 2.1** (Leadership). Within a set of $\mathcal{N}$ workers, each worker generates a prediction error denoted as $\mathcal{L}_l(\vec{y}, \hat{y})$. From this set, we select $\delta$ *leaders* based on their lowest prediction errors. The prediction loss for these leaders is represented as $\mathcal{L}_l^\delta(\vec{y}, \hat{y})$. The remaining workers are referred to as *followers*, and their prediction loss is denoted as $\mathcal{L}_l^{\bar{\delta}}(\vec{y}, \hat{y})$.

*Implementation details.* To enable workers in hidden layers to generate valid predictions, we apply the same activation function used in the output layer to each worker. For instance, in the case of a NN designed for $N$-class classification, we typically include $N$ output neurons in the output layer and apply the softmax function. In our LFNN, each worker is composed of $N$ neurons, and the softmax function is applied accordingly. The leader loss ($\mathcal{L}_l^\delta$) for each output layer is the cross-entropy loss between the outputs and the true labels. In order to align the followers with the leaders, we adopt a simplified approach by selecting the best-performing leader as the reference for computing $\mathcal{L}_l^{\bar{\delta}}$. While other strategies such as random selection from the $\delta$ leaders were also tested, they did not yield satisfactory performance. Therefore, for the sake of simplicity and better performance, we choose the best-performing leader as the reference for the followers' loss computation.

*Practical benefits and overheads.* In contrast to conventional neural networks trained with BP and a global loss, our LFNN-$\ell$ computes worker-wise loss and gradients locally. This approach effectively eliminates backward locking issues, albeit with a slight overhead in local loss calculation. One significant advantage of the BP-free version is that local error signals can be computed in parallel, enabling potential speed-up in the weight update process through parallel implementation.

## 3 EXPERIMENTS

In Section 3.1, we focus on studying the leadership size, conducting an ablation study of loss terms in Eq. 1, and analyzing the worker's activity. To facilitate demonstration and visualization, we utilize DNNs in this subsection. In Section 3.2, we present our main experimental results, where we evaluate LFNNs and LFNN-$\ell$s using CNNs on three datasets (i.e., MNIST, CIFAR-10, and ImageNet) and compare with a set of baseline algorithms. Furthermore, we embed LFNN-$\ell$s within VGG-19 and ResNet-50/101/152 for validation on CIFAR-10, Tiny ImageNet, and ImageNet. This approach aligns with prior experiments conducted in the literature on BP-free algorithms and demonstrates the scalability of our approach Belilovsky et al. (2020); Pyeon et al. (2020).

### 3.1 LEADER-FOLLOWER NEURAL NETWORKS (LFNNS)

**Experimental setup.** To assess the performance of LFNN for online classification, we conduct experiments on the pixel-permuted MNIST dataset LeCun (1998). Following the approach presented in Veness et al. (2019), we construct a one-vs-all classifier using a simple NN architecture consisting of one hidden FC layer. In our experiments, we vary the network architecture to examine the relationship between network performance and leadership size. We consider network configurations with 32, 64, 128, 256, and 512 workers, where each worker corresponds to a single neuron. We systematically vary the percentage of workers assigned as leaders from 10% to 100%. For each network configuration, we utilize the sigmoid activation function for each worker and train the model using the Adam optimizer with a learning rate of 5e-3. The objective is to investigate how different leadership sizes impact the classification performance in the online setting. In our experiments, we

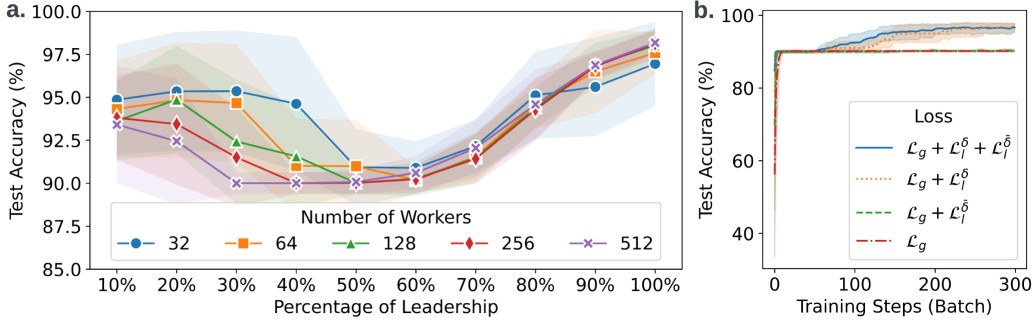

Figure 2: **a.** Network performance results when varying leadership size from 10% to 100%. **b.** Ablation study results from four different loss functions.

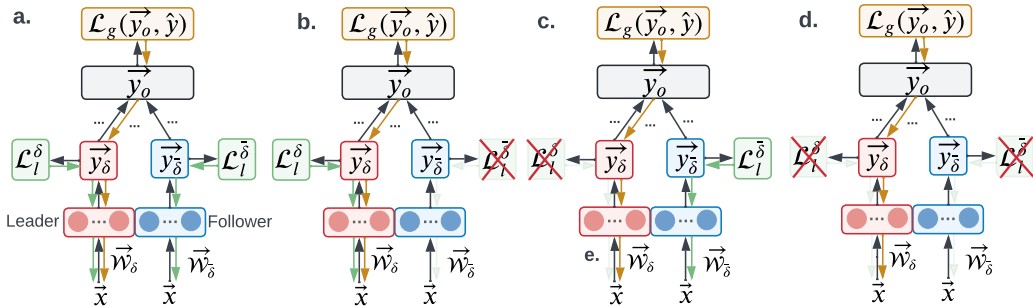

Figure 3: **Loss variation demonstration. a.** Global prediction loss and both local losses, $\mathcal{L}_1$. **b.** Without local follower loss, $\mathcal{L}_2$. **c.** Without local leader loss, $\mathcal{L}_3$. **d.** Global prediction loss alone, $\mathcal{L}_4$.

employ the binary cross-entropy loss for both the global prediction loss ($\mathcal{L}_g$) and the local prediction loss for leaders ($\mathcal{L}_l^\delta$). For the local error signal of followers ($\mathcal{L}_l^{\bar{\delta}}$), we use the mean squared error loss. The hyperparameters $\lambda_1$ and $\lambda_2$ are both set to 1 in this section to balance the global and local loss terms. In the ablation study of loss terms and the worker activity study, we focus on a 32-worker LFNN with 30% leadership.

**Leadership size and performance.** In a study on the collective motion of inanimate objects, such as radio-controlled boats, it was observed that to effectively control the direction of the entire group, only a small percentage (5%-10%) of the boats needed to act as leaders Tarcai et al. (2011). This finding aligns with similar studies conducted on living collectives, such as fish schools and bird flocks, where a small subset of leaders were found to have a substantial impact on the behavior of the larger group. In our experiment, we investigate the relationship between network performance and the size of the leadership group. The results shown in Figure 2a indicate that our LFNN achieves high performance on the permuted MNIST classification task after just one pass of training data. When using a higher percentage of leadership, such as 90% or 100%, the LFNN achieves comparable performance to a DNN trained with BP. Even with a lower percentage of leadership, ranging from 10% to 30%, the LFNN still achieves decent performance on this task. It is worth noting that for more challenging datasets like ImageNet, higher percentages of leadership are preferred. These findings highlight both the similarities and differences between natural collectives and LFNNs in the field of deep learning.

**Ablation study of loss terms.** In our investigation of LFNN training using Eq. 1, we aim to evaluate the effectiveness of the local loss terms and examine the following aspects in this section: (a) whether global loss alone with BP is adequate for training LFNNs, and (b) how the inclusion of local losses contributes to training and network performance in terms of accuracy. To address these questions, we consider four variations of the loss function, as depicted in Figure 3: (i) $\mathcal{L}_1 = \mathcal{L}_g + \mathcal{L}_l^\delta + \mathcal{L}_l^{\bar{\delta}}$: This variant includes the global loss as well as all local losses. (ii) $\mathcal{L}_2 = \mathcal{L}_g + \mathcal{L}_l^\delta$: Here, the global loss is combined with the local leader loss. (iii) $\mathcal{L}_3 = \mathcal{L}_g + \mathcal{L}_l^{\bar{\delta}}$: This variant utilizes the global loss along with the local follower loss. (iv) $\mathcal{L}_4 = \mathcal{L}_g$: In this case, only the global loss is employed.

Figure 4: Leadership in workers during training. The color and size of the dots represent the times a worker is selected as leader. A worker can be selected as a leader up to 300 times in each epoch.

After training LFNNs with the four different loss functions mentioned earlier, we observe the one-pass results in Figure 2b. It is evident that using only the global prediction loss ($\mathcal{L}_4$) with backpropagation leads to the worst performance. The network's accuracy does not improve significantly when adding the local follower loss ($\mathcal{L}_3$) because the leader workers, which the followers rely on for weight updates, do not perform well. As a result, the overall network accuracy remains low. However, when we incorporate the local leader loss ($\mathcal{L}_2$), we notice a significant improvement in the network's performance after 100 training steps. The local leader loss plays a crucial role in this improvement. Despite updating only 30% of the workers at each step, it is sufficient to guide the entire network towards effective learning. Moreover, when we further include the local follower loss ($\mathcal{L}_1$) to update the weights of followers based on strong leaders, the overall network performance improves even further. As a result, the network achieves high accuracy with just one pass of training data. These results highlight the importance of incorporating both local leader and local follower losses in LFNN training. The presence of strong leaders positively influences the performance of followers, leading to improved network accuracy.

**Leadership development.** In order to investigate how leadership is developed during training, we conduct a study using batch training, where leaders are re-selected in each batch. To provide a clearer demonstration, we focus solely on local losses in this study, thereby eliminating the effect of the global error signal and BP. We utilize an LFNN-$\ell$ with two hidden FC layers, each containing 32 workers. The leadership rate is fixed at 20%, resulting in approximately 6 leaders being selected in each layer at every training step. The NN is trained for 300 steps in each epoch, and the visualization of the leadership dynamics during the first 5 epochs is presented in Figure 4. In Figure 4, the visualization depicts the development of leadership during training. Each dot's color and size indicate the number of times a worker is selected as a leader. In the initial epoch (Epoch 0), we observe that several workers in each layer have already emerged as leaders, being selected most of the time. As training progresses, exactly six workers in each layer are consistently developed as leaders, while the remaining workers are no longer selected. By the fifth epoch, the leadership structure becomes nearly fixed, remaining relatively unchanged throughout the training process.

From the results obtained, leadership in LFNN-$\ell$ is developed in the early stages of training and becomes fixed thereafter. The performance of the entire network relies on these leaders. Although this aspect is not the primary focus of the current work, one promising future direction involves the development of an intelligent dynamic leader selection algorithm. Additionally, we also investigated the performance of the best-performing leaders in each layer and compared the performance between leaders and followers in the supplementary materials.

## 3.2 BP-FREE LEADER-FOLLOWER NEURAL NETWORKS (LFNN-$\ell$S)

In this section, we conduct a comparative analysis between LFNN-$\ell$s and several alternative approaches, with the option of engaging BP. We evaluate their performance on the MNIST, CIFAR-10, and ImageNet datasets to showcase the capabilities of LFNN-$\ell$s and further study the impact of leadership size. All LFNN-$\ell$s and LFNNs in this section consist of FC and convolutional layers. LFNNs are trained using a combination of BP, global loss, and local losses, while BP-free LFNN-$\ell$s are trained solely with local losses.

**Datasets.** Both MNIST and CIFAR-10 are obtained from the TensorFlow datasets Abadi et al. (2016). MNIST LeCun (1998) contains 70,000 images, each of size $28 \times 28$. CIFAR-10 Krizhevsky et al. (2009) consists of 60,000 images, each of size $32 \times 32$. ImageNet Deng et al. (2009) contains 1.3 million images of 1000 classes, which we resized to $224 \times 224$. Tiny ImageNet Le & Yang (2015) consists of a dataset of $100,000$ images distributed across 200 classes, with 500 images per class for training, and an additional set of $10,000$ images for testing. All images in the dataset are resized to

| | Dataset | MNIST | MNIST | CIFAR-10 | ImageNet |
|---|---|---|---|---|---|
| | Metric | Test / Train Err. (↓) | Test / Train Err. (↓) | Test / Train Err. (↓) | Test / Train Err. (↓) |
| **BP-enabled** | BP | 2.67 / 0.00 | 2.41 / 0.00 | 33.62 / 0.00 | 36.80 / 14.60 |
| | LG-BP Belilovsky et al. (2019) | 2.43 / 0.00 | 2.81 / 0.00 | 33.84 / 0.05 | 54.37 / 39.66 |
| | **LFNN** | 1.18 / 1.15 | 2.14 / 1.49 | 19.21 / 3.57 | 57.75 / 20.94 |
| **BP-free** | FA Lillicrap et al. (2016) | 2.82 / 0.00 | 2.90 / 0.00 | 39.94 / 28.44 | 94.55 / 94.13 |
| | FG-W Baydin et al. (2022) | 9.25 / 8.93 | 8.56 / 8.64 | 55.95 / 54.28 | 97.71 / 97.58 |
| | FG-A Ren et al. (2022) | 3.24 / 1.53 | 3.76 / 1.75 | 59.72 / 41.27 | 98.83 / 98.80 |
| | LG-FG-W Ren et al. (2022) | 9.25 / 8.93 | 5.66 / 4.59 | 52.70 / 51.71 | 97.39 / 97.29 |
| | LG-FG-A Ren et al. (2022) | 3.24 / 1.53 | 2.55 / 0.00 | 30.68 / 19.39 | 58.37 / 44.86 |
| | **LFNN-ℓ** | **1.49** / 0.04 | **1.20** / 1.15 | **20.95** / 4.69 | **55.88** / 36.13 |
| **Number of Parameters** | | 272K∼275K | 429K∼438K | 876K∼919K | 17.3M∼18.3M |

Table 1: Comparison between the proposed model and a set of BP-enabled and BP-free algorithms under MNIST, CIFAR-10, and ImageNet. The best test errors (%) are highlighted in **bold**. Leadership size is set to 70% for all the LFNNs and LFNN-ℓs.

| Dataset | Model | | Leadership Percentage | | | | | | | | | |
|---|---|---|---|---|---|---|---|---|---|---|---|---|
| | | | 10% | 20% | 30% | 40% | 50% | 60% | 70% | 80% | 90% | 100% |
| **Tiny ImageNet** | LFNN-ℓ | Test | 73.98 | 63.09 | 54.24 | 49.63 | 44.87 | 40.96 | 37.17 | 38.05 | **36.06** | 39.56 |
| | | Train | 71.47 | 57.29 | 43.69 | 38.57 | 30.53 | 22.04 | 19.50 | 19.38 | 16.00 | 32.33 |
| | LFNN | Test | 39.85 | 40.12 | 39.34 | 39.18 | 39.33 | 39.41 | 39.42 | 38.63 | 35.21 | 39.56 |
| | | Train | 36.50 | 35.76 | 32.71 | 32.16 | 32.02 | 32.36 | 32.70 | 31.91 | 32.59 | 32.33 |
| **ImageNet Subset** | LFNN-ℓ | Test | 90.57 | 84.83 | 78.75 | 73.65 | 68.61 | 64.25 | 59.53 | 56.54 | **53.82** | 54.44 |
| | | Train | 68.96 | 51.89 | 39.49 | 27.78 | 22.68 | 13.37 | 9.23 | 5.41 | 5.58 | 6.40 |
| | LFNN | Test | 79.37 | 78.83 | 69.87 | 61.80 | 60.05 | 59.10 | 57.46 | 58.01 | 57.37 | 57.75 |
| | | Train | 53.13 | 52.18 | 38.38 | 26.26 | 25.21 | 20.35 | 18.42 | 18.40 | 16.70 | 17.94 |

Table 2: Error rate (% ↓) results of LFNNs and LFNN-ℓs (with different leadership percentage) on Tiny ImageNet and ImageNet subset. We also trained CNN counterparts (without LF hierarchy) with BP and global loss for reference. The test error rates of BP-enabled CNNs under Tiny ImageNet and ImageNet subset are 35.76% and 51.62%, respectively.

$64 \times 64$ pixels. ImageNet subset (1pct) Chen et al. (2020) is a subset of ImageNet Deng et al. (2009). It shares the same validation set as ImageNet and includes a total of 12,811 images sampled from the ImageNet. These images are resized to $224 \times 224$ pixels for training.

**MNIST and CIFAR-10.** We compare our LFNNs and LFNN-ℓs with BP, local greedy backdrop (LG-BP) Belilovsky et al. (2019), Feedback Alignment (FA) Lillicrap et al. (2016), weight-perturbed forward gradient (FG-W) Baydin et al. (2022), activity perturbation forward gradient (FG-A) Ren et al. (2022), local greedy forward gradient weight / activity-perturbed (LG-FG-W and LG-FG-A) Ren et al. (2022) on MNIST, CIFAR-10, and ImageNet datasets. To ensure a fair comparison, we make slight modifications to our model architectures to match the number of parameters of the models presented in Ren et al. (2022).

Table 1 presents the image classification results for the MNIST and CIFAR-10 datasets using various BP and BP-free algorithms. The table displays the test and train errors as percentages for each dataset and network size. When comparing to BP-enabled algorithms, LFNN shows similar performance to standard BP algorithms and outperforms the LG-BP algorithm on both the MNIST and CIFAR-10 datasets. In the case of BP-free algorithms, LFNN-ℓ achieves lower test errors for both MNIST and CIFAR-10 datasets. Specifically, in MNIST, our LFNN-ℓ achieves test error rates of 2.04% and 1.20%, whereas the best-performing baseline models achieve 2.82% and 2.55%, respectively. For the CIFAR-10 dataset, LFNN-ℓ outperforms all other BP-free algorithms with a test error rate of 20.95%, representing a significant improvement compared to the best-performing LG-FG-A algorithm, which achieves a test error rate of 30.68%[2].

**Scaling up to ImageNet.** Traditional BP-free algorithms have shown limited scalability when applied to larger datasets such as ImageNet Bartunov et al. (2018). To assess the scalability of LFNN and LFNN-ℓ, we conduct experiments on ImageNet subset and Tiny ImageNet[3]. The results in Table 1 compare the train / test error rates of LFNN and LFNN-ℓ with other baseline models using BP and BP-free algorithms on the ImageNet dataset. In the ImageNet experiments, LFNN achieves competitive

---
[2]More experimental results for MNIST and CIFAR-10 under different percentage of leadership can be found in the supplementary materials.
[3]More ImageNet results can be found in the supplementary materials.

test errors compared to BP and LG-BP, achieving a test error rate of 57.75% compared to 36.80% and 54.37% respectively. Notably, when compared to BP-free algorithms, LFNN-$\ell$ outperforms all baseline models and achieves a test error rate 2.49% lower than the best-performing LG-FG-A. Furthermore, LFNN-$\ell$ demonstrates an improvement over LFNN on ImageNet. These results suggest that the use of local loss in LFNN-$\ell$ yields better performance compared to global loss, particularly when dealing with challenging tasks such as ImageNet.

To further investigate the generalizability of LFNN and LFNN-$\ell$, we conduct experiments on ImageNet variants and increase the model size by doubling the number of parameters to approximately 18M. Additionally, we explore the impact of leadership size on model performance. The results of the error rates for Tiny ImageNet and ImageNet subset with varying leadership percentages are presented in Table 2. For Tiny ImageNet, we observe that using a leadership percentage of 90% yields the lowest test error rates, with LFNN achieving 35.21% and LFNN-$\ell$ achieving 36.06%. These results are surprisingly comparable to other BP-enabled deep learning models tested on Tiny ImageNet, such as UPANets (test error rate = 32.33%) Tseng et al. (2022), PreActRest (test error rate = 36.52%) Kim et al. (2020), DLME (test error rate = 55.10%) Zang et al. (2022), and MMA (test error rate = 35.59%) Konstantinidis et al. (2022).

In the ImageNet subset experiments, we follow the methodology of Chen et al. (2020) and leverage the ResNet-50 architecture as the base encoder, combining it with LFNN and LFNN-$\ell$. LFNN and LFNN-$\ell$ with 90% leadership achieve the lowest test error rates of 57.37% and 53.82%, respectively. These results surpass all baseline models in Table 1 and are even comparable to the test error rates of BP-enabled algorithms reported in Chen et al. (2020), which is 50.60%. This observation further demonstrates the effectiveness of our proposed algorithm in transfer learning scenarios.

Furthermore, we observed even better results than those in Table 2 when further increasing the number of parameters. From Figure 2a and Table 5, we recall that for simple tasks like MNIST or CIFAR-10 classification, small leadership sizes can achieve decent results. In Table 2, we observe a clearer trend that for difficult datasets like ImageNet subset, a higher leadership percentage is required to achieve better results. This presents an interesting avenue for future exploration, particularly in understanding the relationship between network / leadership size and dataset complexity.

**Embedding LFNN-$\ell$ in VGG and ResNet**. To evaluate the scalability of LFNN-$\ell$ within the context of classic CNN architectures, we integrated LFNN-$\ell$ into VGG19, ResNet-50, ResNet-101, and ResNet-152. We assess its impact on both accuracy performance (measured by error rates) and speedup performance across various datasets, including CIFAR-10, Tiny-ImageNet, and ImageNet. We compare with two state-of-the-art BP-free block-wise learning algorithms that utilize BP-free optimization: DGL Belilovsky et al. (2020) and SEDONA Pyeon et al. (2020). To ensure a fair comparison, LFNN-$\ell$ employed a leadership size of 100%, guaranteeing that inner communication among leaders and followers would not affect the speedup (the entire architecture will be divided into $K$ blocks of workers). Additionally, LFNN-$\ell$ was configured with the same model implementations and hyperparameter settings (a batch size of 256 using SGD) as DGL and SEDONA[4].

Table 3 (a) and (b) show the prediction error rates for all block-wise BP-free learning models with 4 blocks on CIFAR-10 and Tiny-ImageNet. In almost all cases, DGL performs worse than BP, and SEDONA can marginally outperform BP. In contrast, our LFNN-$\ell$ not only significantly outperforms BP but also outperforms SEDONA and DGL in all cases. Figure 5 illustrates the error rates of various models with different numbers of blocks ($K$). DGL consistently performs worse than BP across all values of $K$. SEDONA's results are similar to BP under Tiny-ImageNet (ResNet-101) and CIFAR-10 (ResNet-152) when $K \geq 8$. In contrast, LFNN-$\ell$ significantly outperforms BP, SEDONA, and DGL in all cases, suggesting that LFNN-$\ell$ exhibits superior scalability compared to SEDONA and DGL. Table 3 (c) and (d) reveal the classification errors on the validation set and training speedup of ImageNet. It is evident that LFNN-$\ell$ outperforms BP, DGL, and SEDONA in terms of both top-1 and top-5 validation errors.

Notably, LFNN-$\ell$ and DGL achieve the highest speedup in ResNet-101 and ResNet-152, respectively. This can be attributed to DGL's uniform network splitting, which significantly enhances parallelization in larger architectures. Speedup is calculated as the ratio of BP's wall-clock training time compared to BP-free benchmarks. To ensure fairness, we calculate all models' speedup values by distributing blocks across $K$ GPUs. Furthermore, LFNN-$\ell$ requires the fewest additional parameters (Params.)

---

[4]DGL: https://github.com/eugenium/DGL; SEDONA: https://github.com/mjpyeon/sedona.

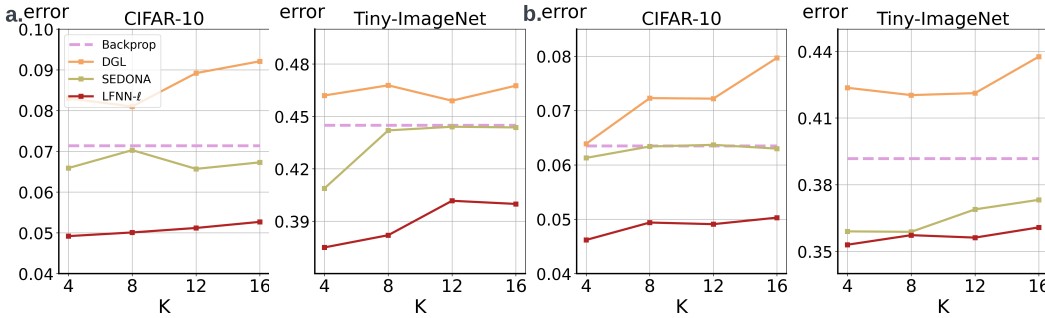

Figure 5: Classification error of **a.** ResNet-101 and **b.** ResNet-152 on CIFAR-10 and Tiny-ImageNet.

for integrating into ResNet-101/152 when compared to other BP-free approaches. The reason for this is that LFNN-$\ell$ does not rely on large auxiliary networks after each block, as SEDONA and DGL do. Of note, unlike the results shown in Table 1, the LFNN-$\ell$ embedded in ResNet exhibits superior performance compared to BP on ImageNet, , indicating that the network size is a key factor in LFNN-$\ell$'s performance. For more complex datasets, LFNN-$\ell$ relies on additional leader-provided information, whereas in small CNN architectures, leaders are unable to gather sufficient information to effectively guide the training process.

| Model | BP | DGL | SEDONA | LFNN-$\ell$ |
|---|---|---|---|---|
| VGG-19 | 12.31 | 12.19 | 11.58 | **6.52** |
| ResNet50 | 7.99 | 8.27 | 7.53 | **5.05** |
| ResNet101 | 7.14 | 8.30 | 6.59 | **4.92** |
| ResNet152 | 6.35 | 6.39 | 6.13 | **4.62** |

(a) CIFAR-10

| Model | BP | DGL | SEDONA | LFNN-$\ell$ |
|---|---|---|---|---|
| VGG-19 | 47.11 | 48.70 | 43.44 | **40.09** |
| ResNet50 | 46.54 | 46.04 | 45.60 | **36.83** |
| ResNet101 | 44.50 | 46.20 | 40.88 | **35.28** |
| ResNet152 | 39.18 | 42.36 | 35.90 | **35.01** |

(b) Tiny-ImageNet

| Method | Top-1 Err. | Top-5 Err. | Speedup Ratios | Params. (M) |
|---|---|---|---|---|
| BP | 21.62 | 5.94 | 1 | **44.55** |
| DGL | 22.35 | 6.44 | 1.92 | 47.09 |
| *SEDONA | 21.00 | 5.52 | 2.01 | 70.36 |
| LFNN-$\ell$ | **20.92** | **5.44** | **2.07** | 46.34 |

*Results are from SEDONA.

(c) ResNet-101

| Method | Top-1 Err. | Top-5 Err. | Speedup Ratios | Params. (M) |
|---|---|---|---|---|
| BP | 21.40 | 5.69 | 1 | **60.19** |
| DGL | 22.20 | 6.39 | **2.23** | 62.73 |
| *SEDONA | 20.20 | 5.13 | 2.02 | 86.00 |
| LFNN-$\ell$ | **20.08** | **5.01** | 2.21 | 61.98 |

*Results are from SEDONA.

(d) ResNet-152

Table 3: Error rates (% ↓) on CIFAR-10 (a) and Tiny-ImageNet (b) with different methods with 4 blocks. Error rates (% ↓), speedup ratios (↑), and the number of parameters (↓) are compared among different methods on ResNet-101 (c) and -152 (d), each with 4 blocks, when applied to ImageNet.

## 4 CONCLUSION

In this work, we have presented LFNN, inspired by collective behavior observed in nature. By introducing a leader-follower hierarchy within NNs, we have demonstrated its effectiveness across various network architectures. Our comprehensive study of LFNN aligns with observations and theoretical foundations in both the biological and deep learning domains. In addition, we have proposed LFNN-$\ell$, a BP-free variant that utilizes local error signals instead of traditional backpropagation. We have shown that LFNN-$\ell$, trained without a global loss, achieves superior performance compared to a set of BP-free algorithms. Through extensive experiments on MNIST, CIFAR-10, and ImageNet datasets, we have validated the efficacy of LFNN with and without BP. LFNN-$\ell$ not only outperforms other state-of-the-art BP-free algorithms on all tested datasets but also achieves competitive results when compared to BP-enabled baselines in a considerable amount of cases. Furthermore, LFNN-$\ell$ can be integrated into convolutional neural networks such as VGG and ResNet architectures, which significantly accelerates the training process. Our work is the first to introduce collective motion-inspired models for deep learning architectures, opening up new directions for the development of local error signals and alternatives to BP. The proposed algorithm is straightforward yet highly effective, holding potential for practical applications across various domains. We believe this early study offers valuable insights into fundamental challenges in deep learning, including NN architecture design and the development of biologically plausible decentralized learning algorithms.

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

## A    CODE AND DATA AVAILABILITY

Our code to run the experiments can be find at `https://anonymous.4open.science/r/LFNN-6DF4/`.

## B    RELATED WORK

Efforts have been made to bridge the gaps in computational efficiency that continue to exist between ANNs and BNNs Bartunov et al. (2018). One popular approach is the replacement of global loss with local error signals Mostafa et al. (2018). Researchers have proposed to remove BP to address backward locking problems Jaderberg et al. (2017), mimic the local connection properties of neuronal networks Pyeon et al. (2020) and incorporate local plasticity rules to enhance ANN's biological plausibility Illing et al. (2021). A research topic closely related to our work is supervised deep learning with local loss. It has been noticed that training NNs with BP is biologically implausible because BNNs in the human brain do not transmit error signals at a global scale Crick et al. (1989); Marblestone et al. (2016); Lillicrap et al. (2020). Several studies have proposed training NNs with local error signals, such as layer-wise learning Mostafa et al. (2018); Nøkland & Eidnes (2019), block-wise learning Pyeon et al. (2020); Löwe et al. (2019), gated linear network family Veness et al. (2019), etc. Mostafa et al. generate local error signals in each NN layer using fixed, random auxiliary classifiers Mostafa et al. (2018), where a hidden layer is trained using local errors generated by a random fixed classifier. This is similar to an approach called feedback alignment training, where random fixed weights are used to back-propagate the error layer by layer Lillicrap et al. (2016). In Löwe et al. (2019), the authors split a NN into a stack of gradient-isolated modules, and each module is trained to maximally preserve the information of its inputs. A more recent work by Ren et al. Ren et al. (2022) proposed a local greedy forward gradient algorithm by enabling the use of forward gradient learning in supervised deep learning tasks. Their biologically plausible BP-free algorithm outperforms the forward gradient and feedback alignment family of algorithms significantly. Our LFNN-$\ell$ shares some similarities with the above work in the sense that the LFNN-$\ell$ is trained with loss signals generated locally without BP. In contradistinction to the state-of-the-art, we do not require extra memory blocks to generate an error signal. Hence, the number of trainable parameters can be kept identical to that of NNs without an LF hierarchy.

## C    MORE EXPERIMENTS AND DICUSSION

### C.1    MNIST

**Experimental setup.** In this section, we follow Section 3.1 to conduct experiments with a LFNN-$\ell$ for MNIST data classification. We utilize a network consisting of two hidden fully connected layers, each containing 32 workers. Given that it is a 10-class classification problem, each worker is naturally equipped with 10 neurons, as specified in the implementation details outlined in the methods. We vary the leadership size from 10% to 100% and compare the BP-free version with its BP-enabled counterpart. The *softmax* and ReLU activation functions are used for the output and hidden layers, respectively. An Adam optimizer with a learning rate of 5e-4 is employed, and the hyper-parameter $\lambda$ is set to 1. Categorical cross-entropy loss is utilized as the prediction loss for output neurons ($\mathcal{L}_g^o$) and leader workers ($\mathcal{L}_l^\delta$), while mean squared error is employed as a local error signal for follower workers ($\mathcal{L}_l^{\bar{\delta}}$). To train the networks, we conduct 50 epochs with early stopping criteria. The results presented in Table 4 are evaluated based on 5 repetitions of the experiment using different random initializations. In addition to comparing BP-enabled and BP-free LFNNs, we examine the performance of leaders and provide a visualization of their development. Furthermore, we investigate and compare the abilities of leaders and workers. For these analyses, we utilize a trained LFNN-$\ell$ model with a leadership size of 20%.

**Experimental results.** Based on the comparison results presented in Table 4, we observe that both BP-free and BP-enabled LFNNs yield similar results, albeit slightly lower than a classical DNN[5]. However, a notable difference arises when adjusting the leadership size. Specifically, as the leadership

---

[5]A MLP trained with BP and possessing the same number of trainable parameters achieves a test accuracy of 98.11%.

| Leadership | 10% | 20% | 30% | 40% | 50% | 60% | 70% | 80% | 90% | 100% |
|---|---|---|---|---|---|---|---|---|---|---|
| BP-free | 96.40% | **96.68%** | 96.23% | 96.17% | 95.98% | 95.89% | 95.93% | 95.39% | 95.22% | 94.87% |
| BP-enabled | 96.08% | 96.30% | 96.41% | 96.38% | **96.52%** | 96.19% | 96.21% | 96.34% | 96.29% | 96.20% |

Table 4: Accuracy results of an LFNN trained with and without BP.

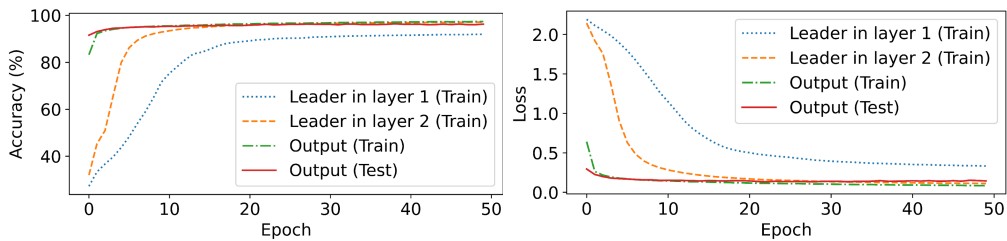

Figure 6: **Leader accuracy and loss analysis.** The best-performing leaders in hidden layers generate accurate predictions. Leaders closer to the output layer perform better than leaders in early layers.

size increases, the test accuracy of LFNN-$\ell$ tends to decrease, whereas the accuracy of the BP-enabled version remains relatively stable. This discrepancy can be attributed to the potency of the global prediction loss, which is known to be highly effective in conjunction with BP. Therefore, given a sufficient training duration, the network is capable of achieving comparable accuracies as long as an adequate number of leader workers are present. Increasing the number of leaders does not confer additional benefits to the overall network performance for this simple task.

In contrast, the BP-free network does not rely on the propagation of a global loss signal through the layers to adjust weights. Instead, it relies on local prediction losses to update leader workers. In nature, a small number of participants often suffices to dominate and guide the movement of a group. In simple scenarios, a large leadership size does not necessarily contribute to higher accuracy or ensure the correct direction of the entire group. In fact, it may even impede performance, as leaders receive local signals that may not be aligned with one another. In LFNN, however, leaders receive global loss signals without potential conflicts in the error signal. Therefore, larger leadership sizes do not compromise overall accuracy.

**The best-performing leaders.** In order to gain a deeper understanding of the LFNN's leadership dynamics, we examine the performance of the leaders that the followers choose to follow during training. As described in Section 2, followers in a hidden layer select the leader with the best performance. However, it is important to note that this approach may have limitations if the best-performing leaders do not themselves perform well, potentially impacting the overall network performance. To investigate this further, we analyze the train and test accuracies of the final output layer throughout the training process. From the results depicted in Figure 6, we observe that the accuracy of the output layer reaches a high level early on during training. Additionally, the accuracy of leaders in hidden layers gradually improves as training progresses. Notably, leaders in layers closer to the output layer tend to outperform leaders located in layers closer to the input layer. This observation suggests that information flows effectively through the network, and leaders in later layers contribute more significantly to the overall performance.

The observations from our analysis yield two key messages. Firstly, the progressive processing and learning of information in the LFNN validate the effectiveness of approaches such as transfer learning or learning with pre-trained models. Specifically, the utilization of representations from later layers, which contain more useful information for classification, aligns with the common practice of using pre-trained models by discarding the last output layer and utilizing the representation generated by the last hidden layer for downstream tasks. Secondly, the performance of the best-performing leaders in each hidden layer highlights their ability to generate accurate predictions. Consequently, the followers aligning their representations with these leaders can be viewed as a form of layer-wise knowledge distillation. In this analogy, the leaders function as teacher models, distilling their knowledge, while the followers act as student models, assimilating the distilled knowledge. This concept of knowledge

distillation has been shown to outperform learning solely from raw labels in certain scenarios Gou et al. (2021). Notably, we find that the LFNN-$\ell$ with 100% leadership, which corresponds to learning solely from raw labels, performs worse than other configurations, as indicated in Table 4.

Overall, these findings highlight the importance of both layer-wise knowledge distillation and the progressive information processing in LFNNs, shedding light on the potential benefits of incorporating pre-trained models and teacher-student learning paradigms in deep learning approaches.

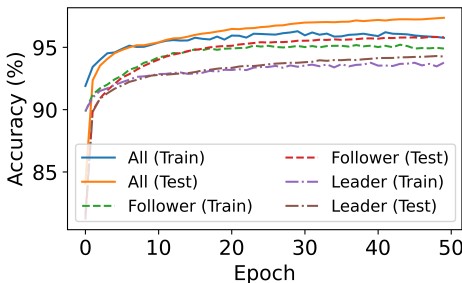

Figure 7: Accuracy results of LFNN-$\ell$ with different workers in inference.

**Leaders vs. followers.** In this section, we investigate the individual performance of leaders and followers in the LFNN architecture during the inference phase. By isolating the leaders and followers separately, we aim to understand their contributions and evaluate their effectiveness in generating accurate predictions. We create two ablated variants: one with only leader workers in inference and another with only follower workers. It is important to note that using only a subset of workers may impact the efficiency of the network since a portion of the neurons remain unused. The accuracy results, as depicted in Figure 7, demonstrate that the network achieves the highest accuracy when both leader and follower workers are employed in the inference process. Interestingly, we observe that the network with follower workers alone in inference outperforms the network with only leader workers. This finding aligns with our earlier discussion regarding the roles of leaders and followers. In our LFNN setup, leaders act as teacher models, while followers serve as student models. Traditionally, in knowledge distillation, teacher models tend to outperform student models due to their larger and more complex architectures. However, in our specific setup, the follower workers learn from the best-performing teacher, which explains their superior performance. These results highlight the significance of incorporating both leaders and followers in the LFNN architecture during inference. While followers benefit from the knowledge distilled by the leaders, the presence of leader workers enhances the overall performance of the network. This finding underscores the importance of the dynamic interaction between leaders and followers, and their collective contribution to achieving high accuracy in the LFNN framework.

**Worker activity in an LFNN.** Collective motion in a group of particles can be readily identified through visualization. Since our LFNN's weight update rules are inspired by a collective motion model, we visualize the worker activities and explore the existence of collective motion patterns in the network during training. Following our weight update rule, we select 30% of the leaders from the 32 workers in each training step and update their weight dynamics based on global and local prediction loss. Consequently, the leader workers receive individual error signals and update their activity accordingly. Conversely, the remaining 70% of workers act as followers and update their weight dynamics by mimicking the best-performing leader through local error signals. In essence, all followers align themselves with a single leader, resulting in similar and patterned activity in each training step.

To visualize the activities of all workers, we utilize the neuron output $\vec{y}$ before and after the weight update at each time step, and the difference between them represents the worker activity. The results in Figure 8a demonstrate that in each time step, the follower workers (represented by blue lines) move in unison to align themselves with the leaders. During the initial training period (steps 0 to 1000), both leaders and followers exhibit significant movement and rapid learning, resulting in relatively larger step sizes. As the learning process stabilizes and approaches saturation, the workers' movement becomes less pronounced as the weights undergo less drastic changes in the well-learned network. Overall, we observe a patterned movement in worker activity in LFNNs, akin to the collective motion observed in the classic Vicsek model Vicsek et al. (1995).

**Leadership size on training/testing MNIST and CIFAR-10.** Table 5 presents the relationship between leadership size and model performance on validating MNIST and CIFAR-10. In MNIST, LFNN and LFNN-$\ell$ with different leadership sizes achieve similar test error rates. Further details on the relationship between leadership size and model performance will be discussed in the next subsection.

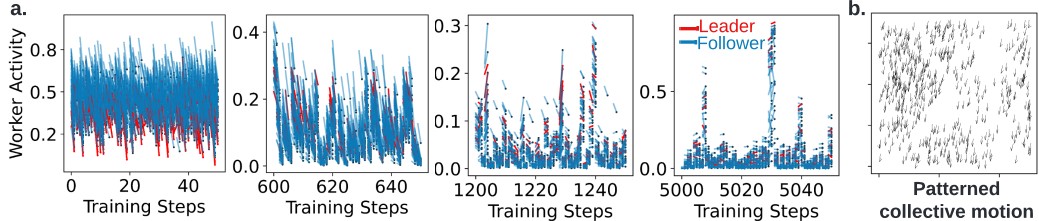

Figure 8: **a.** Worker activity visualization in an LFNN. At each time step, the followers (blue lines) align themselves with leaders (red lines). **b.** Patterned collective motion produced by the classic Vicsek model Vicsek et al. (1995).

| Dataset (No. of Parameters) | Model | | Leadership Percentage | | | | | | | | | |
|---|---|---|---|---|---|---|---|---|---|---|---|---|
| | | | 10% | 20% | 30% | 40% | 50% | 60% | 70% | 80% | 90% | 100% |
| **MNIST** (∼275K) | LFNN-ℓ | Test | 1.96 | 1.67 | 1.98 | 1.49 | 1.49 | 1.49 | **1.49** | 1.64 | 1.69 | 1.57 |
| | | Train | 0.12 | 0.42 | 0.07 | 0.06 | 0.05 | 0.11 | 0.04 | 0.36 | 0.24 | 0.92 |
| | LFNN | Test | 1.24 | 1.68 | 1.50 | 1.60 | 1.40 | 1.20 | **1.18** | 1.40 | 1.44 | 1.51 |
| | | Train | 1.21 | 1.21 | 1.20 | 1.20 | 1.10 | 1.10 | 1.15 | 1.10 | 1.21 | 1.17 |
| **MNIST** (∼438K) | LFNN-ℓ | Test | 1.25 | 1.49 | 1.85 | **1.12** | 1.27 | 1.76 | 1.20 | 1.64 | 1.20 | 1.23 |
| | | Train | 1.14 | 1.22 | 1.30 | 1.22 | 1.17 | 1.21 | 1.15 | 1.15 | 1.18 | 1.13 |
| | LFNN | Test | **1.89** | 2.49 | 2.20 | 2.97 | 2.23 | 2.70 | 2.14 | 2.27 | 2.20 | 2.67 |
| | | Train | 1.70 | 2.17 | 2.08 | 1.84 | 2.06 | 1.97 | 1.49 | 1.93 | 1.61 | 1.58 |
| **CIFAR-10** (∼876K) | LFNN-ℓ | Test | 23.37 | 23.09 | 21.26 | 21.56 | 21.11 | 21.57 | **20.95** | 21.21 | 21.28 | 21.34 |
| | | Train | 6.20 | 5.65 | 3.57 | 3.85 | 4.07 | 4.20 | 4.69 | 5.09 | 4.38 | 4.47 |
| | LFNN | Test | 23.36 | 20.11 | 19.56 | 19.32 | 19.64 | 18.84 | 19.21 | 19.84 | 19.92 | **18.41** |
| | | Train | 8.59 | 5.56 | 3.63 | 4.05 | 4.31 | 4.89 | 3.57 | 5.43 | 3.06 | 3.37 |

Table 5: Error rate (% ↓) results of LFNNs and LFNN-ℓs (with different leadership percentage) on MNIST and CIFAR-10.

| Dataset (No. of Parameters) | Model | | Leadership Percentage | | | | | | | | | |
|---|---|---|---|---|---|---|---|---|---|---|---|---|
| | | | 10% | 20% | 30% | 40% | 50% | 60% | 70% | 80% | 90% | 100% |
| **ImageNet** (∼18.3M) | LFNN-ℓ | Test | 95.21 | 95.80 | 68.22 | 59.43 | 59.95 | 55.58 | 55.88 | 51.59 | 47.76 | **43.38** |
| | | Train | 94.83 | 94.09 | 57.69 | 52.67 | 54.31 | 38.84 | 36.13 | 32.45 | 31.70 | 29.66 |
| | LFNN | Test | 86.51 | 67.88 | 59.20 | 55.75 | 58.27 | 57.01 | 57.75 | 53.66 | 48.18 | **44.34** |
| | | Train | 84.74 | 61.90 | 54.49 | 32.96 | 20.75 | 29.24 | 20.94 | 20.26 | 23.09 | 20.23 |

Table 6: Error rate (% ↓) results of LFNNs and LFNN-ℓs (with different leadership percentage) on ImageNet.

## C.2 IMAGENET

**Experimental setup.** In this section, we conducted experiments to evaluate the performance of LFNN-ℓ and LFNN on the ImageNet dataset Deng et al. (2009). The aim was to investigate the relationship between network performance and leadership size. Following the methodology described in Section 3.2, we trained NNs with approximately 18.3 million trainable parameters. The models were trained using the Adam optimizer with a learning rate of 1e-3 and employed ReLU and Softmax activation functions. All experiments were conducted using 4 Nvidia A100 GPUs. For the loss function, we utilized sparse cross-entropy loss for both the global prediction loss ($\mathcal{L}_g$) and the local prediction loss for leaders ($\mathcal{L}_l^{\delta}$). The local error signal for followers ($\mathcal{L}_l^{\bar{\delta}}$) was measured using the mean squared error loss. In this section, we set the hyperparameters $\lambda_1$ and $\lambda_2$ to 1 to balance the global and local loss terms for LFNN, and $\lambda$ was set to 1 for LFNN-ℓ.

**Dataset.** ImageNet Deng et al. (2009) consists of 1.3 million images belonging to 1000 classes. The images were resized to dimensions of $224 \times 224$. Note that we did not apply any data augmentation or random crop techniques in this study.

**Experimental results.** We present the detailed test and train error rates of LFNN-ℓ and LFNN on the ImageNet dataset with varying leadership percentages, as shown in Table 6. It can be observed that for both LFNN-ℓ and LFNN, the best performance is achieved with a leadership percentage of

| Dataset | Model | $\lambda$ | | | | | | | | | |
|---------|-------|------|------|------|------|------|------|------|------|------|------|
| | | 0.2 | 0.4 | 0.6 | 0.8 | 1.0 | 1.2 | 1.4 | 1.6 | 1.8 | 2.0 |
| **MNIST** | LFNN-$\ell$ | 1.51 | 1.46 | 1.48 | 1.40 | **1.20** | 1.40 | 1.41 | 1.37 | 1.40 | 1.35 |
| | LFNN | 1.51 | 1.51 | 1.52 | 1.50 | **1.18** | 1.46 | 1.43 | 1.47 | 1.46 | 1.41 |
| **CIFAR-10** | LFNN-$\ell$ | 22.43 | 23.77 | 21.79 | 21.54 | **20.95** | 23.13 | 21.94 | 22.06 | 22.67 | 21.37 |
| | LFNN | 20.31 | 19.37 | 20.46 | 19.66 | **19.21** | 19.31 | 19.95 | 19.66 | 19.78 | 23.50 |

Table 7: Error rate (% ↓) results of BP-free models (with values of $\lambda$) on MNIST and CIFAR-10.

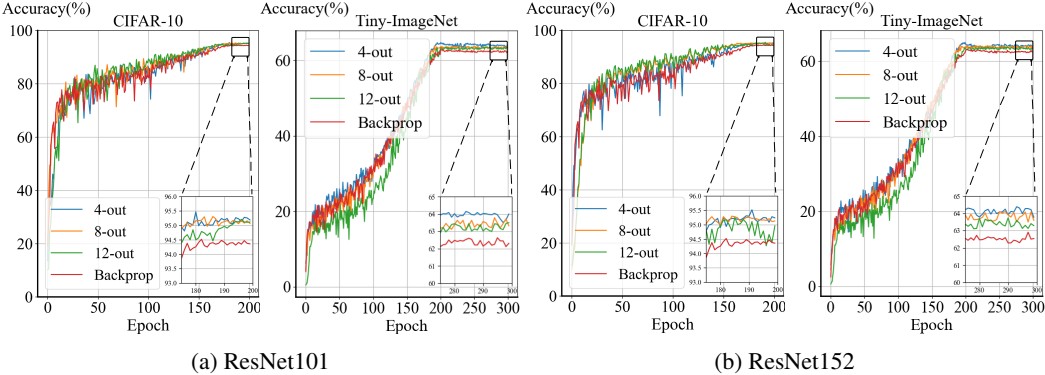

(a) ResNet101         (b) ResNet152

Figure 9: Classification accuracy curves of (a) ResNet-101 and (b) -152 with increasing $K$ and end-to-end BP on CIFAR-10 and Tiny-ImageNet.

100%, resulting in error rates of 49.38% and 46.36%, respectively. This observation suggests that in complex datasets such as ImageNet, a larger leadership size is beneficial for achieving better results. This finding aligns with our conclusion in the main text, further demonstrating the generalization ability of our proposed LFNN(-$\ell$) approach to larger and more challenging datasets and tasks.

### C.3 ABLATION STUDY FOR $\lambda$

The ablation study results of $\lambda$ for LFNN-$\ell$ is presented in Table 7. From the results, we can notice that all the $\lambda$ values have the comparable error rates. Notably, $\lambda = 1$ consistently delivers the best performance across all methods. Therefore, for consistency, we opt for $\lambda = 1$ in all our experiments.

## D EMBEDDING LFNN-$\ell$ IN VGG/RESNET

### D.1 BASED ARCHITECTURES AND COMPUTING ENVIRONMENTS

Following the setup in Pyeon et al. (2020), we remove the MaxPool layer after the initial convolutional layer in ResNet variants for validating CIFAR-10 and Tiny-ImageNet. All experiments are conducted with 4 NVIDIA A100 80 GB GPU cards.

### D.2 OPTIMIZATION

For a fair comparison, we follow the same training setup in Pyeon et al. (2020).

**CIFAR-10.** In all CIFAR-10 experiments, we train models for 64,000 iterations, using an SGD optimizer with a batch size of 128 and a momentum of 0.9. The learning rate is initially set to 0.1 and gradually decays to 0.001. Weight decay settings differ based on the architecture: 0.0001 for VGG-19 and ResNet-50/101, and 0.0002 for ResNet-152, respectively.

**Tiny-ImageNet.** In all Tiny-ImageNet experiments, we train models for 30,000 iterations, using an SGD optimizer with a batch size of 256 and a momentum of 0.9. The initial learning rate is set to 0.1 and is reduced to 0.001 using a cosine annealing schedule. Weight decay is applied at 0.0001 for VGG-19 and ResNet-50/101, and at 0.0002 for ResNet-152, respectively.

**ImageNet.** In all ImageNet experiments, models undergo training for 600,000 iterations with a batch size of 256, employing an SGD optimizer with a momentum of 0.9. Weight decay is set at 0.00005 for both ResNet-101 and ResNet-152. The initial learning rate is 0.1 for ResNet-101, which decays to 0.0001, and 0.05 for ResNet-152, which decays to 0.00001.

### D.3    CONVERGENCE OF LFNN-$\ell$ AND BP

The classification accuracy curves of LFNN-$\ell$ with varying numbers of blocks and BP on CIFAR-10 and Tiny-Imagenet are depicted in Figure 9. Panels (a) and (b) correspond to ResNet-101 and -152, respectively. Notably, in both CIFAR-10 and ImageNet, LFNN-$\ell$ with all numbers of blocks exhibits a similar convergence speed as BP.

