# OpenReview forum: "Enhancing Neural Network Performance with Leader-Follower Architecture and Local Error Signals"
_ICLR.cc/2024/Conference — ICLR 2024 Conference Withdrawn Submission_

### Official Review · Reviewer_2vx7 · 2023-11-01

**Soundness:** 3 good
**Presentation:** 2 fair
**Contribution:** 3 good
**Rating:** 6
**Confidence:** 3

**Summary:**

This paper introduces the concept of workers that encompass one or more information
processing units, which can be either leader or follower (LFNN) that leveraging local error signals to learn. LFNN does not require backprobagation (BP) and global loss to achieve the best performance. Extensive experiments are performed to verify the proposed method.

**Strengths:**

1. The design of leader-follower workers that performs collaborative learning in a neural network is very interesting.

2. Experimental results seems to be promising.

**Weaknesses:**

1. Although the proposed method is claimed to be inspired from biological neural network, the rationale behind it is still unclear to me. I’m not an expert in biological neural network, but I assume any improvement of the new design should have theoretical-grounded explanation. Could the author of the paper provide such theoretical proof on the proposed method?

**Questions:**

Please refer to weaknesses.

---

> ### Author Response · Authors · 2023-11-16
> **Response to Reviewer 2vx7**
>
> We thank the reviewer for comprehensively assessing our manuscript and we are glad to see the reviewer has acknowledged our work as “very interesting” and “promising”.
>
> Our LFNN and LFNN-$\ell$ methods draw inspiration from collective behavior observed in biological neural networks (BNNs) found in nature, such as cells, birds, and human brains. BNNs showcase self-organizing local connections among nodes, enabling them to learn and process limited or noisy input data [1]. Unlike BNNs, artificial neural networks (ANN) employ a layered structure where all information is analyzed at the end (output layer), this design limitation arises because the layered structure cannot be parallelized, and components in the architecture cannot communicate with each other. Furthermore, in biological networks such as the human brain, synaptic weight updates can occur through local learning, independent of the activities of neurons in other brain regions [2, 3]. Local learning has been identified as effective means to reduce memory usage during training and to facilitate parallelism in deep learning architectures [4]. With local loss in our LFNN-$\ell$, we also observe speedup in the training process.
>
> That’s said, our goal is to draw inspiration from complex collectives and integrate this concept into artificial neural networks (ANNs) by introducing local loss functions which aim to parallelize the computation process during training. We may be not able to provide formulas as a mathematical proof, but we posit that our local loss-based approaches (LFNN) share similarities with BNNs on a higher and more abstract level, contributing to the facilitation of learning and understanding of input data. As a compensation, we show extensive experimental results as practical proof of concept to demonstrate that LFNN-$\ell$ consistently outperformed both BP and other BP-free approaches. We hope this work can open up new directions in ANN design and provide inspiration to the broader NN community, and we hope this response addresses the comment from the reviewer.
>
>
> [1] Henry Markram, Wulfram Gerstner, and Per Jesper Sjöström. A history of spike-timingdependent plasticity. Frontiers in synaptic neuroscience, 3:4, 2011.
>
> [2] Natalia Caporale and Yang Dan. Spike timing–dependent plasticity: a hebbian learning rule. Annu. Rev. Neurosci., 31:25–46, 2008
>
> [3] Chenzhong Yin, Phoebe Imms, Mingxi Cheng, Anar Amgalan, Nahian F Chowdhury, Roy J Massett, Nikhil N Chaudhari, Xinghe Chen, Paul M Thompson, Paul Bogdan, et al. Anatomically interpretable deep learning of brain age captures domain-specific cognitive impairment. Proceedings of the National Academy of Sciences, 120(2):e2214634120, 2023
>
> [4] Deepak Narayanan, Aaron Harlap, Amar Phanishayee, Vivek Seshadri, Nikhil R Devanur, Gregory R Ganger, Phillip B Gibbons, and Matei Zaharia. Pipedream: Generalized pipeline parallelism for dnn training. In Proceedings of the 27th ACM Symposium on Operating Systems Principles , pp. 1–15, 2019.

---

### Official Review · Reviewer_Mhh7 · 2023-11-01

**Soundness:** 3 good
**Presentation:** 3 good
**Contribution:** 3 good
**Rating:** 5
**Confidence:** 3

**Summary:**

The paper introduced the worker concept to neural networks and divided the components of NNs into leaders and followers. By leveraging the local error signals, a novel BP-free training method is proposed. And extensive study shows that the proposed training method can achieve promising results on the benchmark datasets.

**Strengths:**

1. The idea that combines the worker concept with the training of neural network is novel and  interesting.

2. The proposed method is comprehensively evaluated and the performance on the benchmark is promising.

3. The paper is well written and organized.

**Weaknesses:**

1.The proposed method can outperform other BP-free baselines on different benchmark datasets. However, the performance of proposed LFNN method seems not promising on complete ImageNet-1K, compared to BP method. It may suggest that the proposed method may not be effective in large scale datasets.

2.The comparsion of the convergence of BP, other BP-free method and the proposed method is missing in the paper.

3.The ablation study on hyper-parameter $\lambda$ is missing in the paper.

**Questions:**

1.The results in Fig.2a suggest that the models with approximately 50%-60% leaders would achieve lowest performance. Can authors explain why that happen?

2.The proposed method requires 90% leadership on Tiny ImageNet and ImageNet subset and 100% leadership on complete ImageNet for both LFNN and LFNN-$\ell$. Does it mean that the demand for the leadership is increasing as the size of dataset grows? Would that increase the computation cost for the training?

3.Compared to BP and other BP-free methods, is LFNN-$\ell$ more time-efficient during the training?

---

> ### Author Response · Authors · 2023-11-16
> **Response to Reviewer Mhh7**
>
> We value the comprehensive review conducted by the reviewers and express gratitude for acknowledging our work as “novel and interesting”. Below, we provide detailed responses to address each specific comment, ensuring clarity and transparency.
>
> Weaknesses:
>
> (1) We thank the reviewer for the correct observation that our “proposed method can outperform other BP-free baselines on different benchmark datasets”. Scaling to larger datasets is for sure a common challenge for BP-free algorithms [1, 2, 3, 4]. We would like to clarify that our BP-free method (LFNN-$\ell$) actually can outperform all BP-free and BP baselines on larger-scale problems if we increase the complexity of networks. As illustrated in Table 3 (c and d), when applying LFNN-$\ell$ on ResNet, LFNN-$\ell$ demonstrates results that are comparable to those achieved with the BP algorithm. Hence, the principal factor contributing to LFNN's relatively inferior performance shown in Table 1 compared to BP on larger-scale problems appears to be the small network size.
>
> [1] Ren, Mengye, Simon Kornblith, Renjie Liao, and Geoffrey Hinton. "Scaling forward gradient with local losses." arXiv preprint arXiv:2210.03310 (2022).
>
> [2] Timothy P. Lillicrap, Daniel Cownden, Douglas B. Tweed, and Colin J. Akerman. Random synaptic feedback weights support error backpropagation for deep learning. Nature Communications, 7(1): 13276, Nov 2016. ISSN 2041-1723. doi: 10.1038/ncomms13276.
>
> [3] Arild Nøkland. Direct feedback alignment provides learning in deep neural networks. In Advances in Neural Information Processing Systems 29, NeurIPS, 2016.
>
> [4] Sergey Bartunov, Adam Santoro, Blake A. Richards, Luke Marris, Geoffrey E. Hinton, and Timothy P. Lillicrap. Assessing the scalability of biologically-motivated deep learning algorithms and architectures. In Advances in Neural Information Processing Systems 31, NeurIPS, 2018.
>
> (2) We sincerely appreciate the reviewer's insightful comments and constructive feedback, which have significantly contributed to enhancing the comprehensiveness of our work. To address the reviewer's likely interest in observing the convergence of learning curves (test accuracy) during the training process, we have included classification accuracy curves for LFNN-$\ell$ and BP of ResNet-101 and ResNet-152 on CIFAR-10 and Tiny-ImageNet in our revised manuscript (refer to “Convergence of LFNN-$\ell$ and BP” section). These curves can be found in Figure 9 in the revised manuscript (we didn’t generate the curves for other BP-free approaches because of the time and resource limit). “x-out” in the plots represents the different number of blocks that we used in LFNN-$\ell$. Notably, in both CIFAR-10 and ImageNet, LFNN-$\ell$ with all numbers of blocks exhibits a similar convergence speed as BP.
>
> (3) We express our gratitude to the reviewer for providing this comment, which has contributed to improving the comprehensiveness of our experiments. To address this, we conducted an ablation study of lambda for both LFNN and LFNN-$\ell$ on both MNIST and CIFAR-10 datasets, considering values ranging from 0.2 to 2 (lambda equal to 0 is validated in Figure 2 b). The results are shown in the “Ablation study for $\lambda$” and Table 7 in our revised manuscript. Examining the table, we observe that all $\lambda$ values exhibit a similar testing error rate. Notably, $\lambda=1$ consistently delivers the best performance across all methods. Therefore, for consistency, we opt for $\lambda = 1$ in all our experiments.

---

> > ### Author Response · Authors · 2023-11-16
> > **Response to Reviewer Mhh7**
> >
> > Questions
> >
> > (1)  We appreciate the reviewer for bringing up this question. In Figure 2a, we present the test accuracy for LFNN on MNIST after one-pass of training, representing the output results after the first epoch.
> >
> > Given that our LFNN draws inspiration from collective behaviors, and considering that in real-world conditions only 5%-10% of individuals are typically chosen as leaders [1], we formulate the following hypothesis to elucidate this phenomenon: Intuitively, the higher the information leadership size would result in better decision making in complex collectives in nature. However, artificial neural networks are very different from the collective behaviors observed in nature. For more challenging tasks (CIFAR-10, ImageNet), we do observe that higher leadership size tends to give higher accuracy (in Table 5 and 6). In this one-pass training of MNIST, we do not observe this pattern. Our hypothesis is that with few and considerable amounts of leadership, there is less randomness and noisy information during training. With around 50\% leadership, for this small neural network on a simple task with one-pass training, the observation could be caused by the randomness and competing information received by followers.
> >
> > [1] Peter Friedl, Yael Hegerfeldt, and Miriam Tusch. Collective cell migration in morphogenesis and cancer. International Journal of Developmental Biology, 48(5-6):441–449, 2004.
> >
> > (2) We agree with the reviewer's perspective that larger leadership sizes are needed for the increasing complexity of the dataset. Tables 5 and 6 provide a comprehensive view of the error rate results for both LFNN and LFNN-$\ell$ across MNIST, CIFAR-10, and ImageNet, with varying leadership sizes. Upon careful examination, it becomes evident that in both MNIST and CIFAR-10, different leadership sizes yield comparable error rates. However, when considering ImageNet, a notable trend emerges, where models with larger leadership sizes exhibit superior performance. This phenomenon can be attributed to the architecture's need for more information when analyzing complex datasets. This requirement necessitates a higher number of leader workers to effectively guide the training process.
> >
> > In LFNN and LFNN-$\ell$, increasing leadership sizes results in a slight increase in the training computation cost. This is because leader workers in some cases necessitate output neurons to compute local loss, introducing a slight overhead in the number of parameters. However, a higher leadership size corresponds to enhanced performance, as evidenced in Table 5 and 6. Therefore, there exists a minor trade-off between computation cost and performance.
> >
> > (3) We express our gratitude to the reviewer for providing this comment, which has contributed to improving the comprehensiveness of our experiments. To address this, we conducted an ablation study of lambda for both LFNN and LFNN-$\ell$ on both MNIST and CIFAR-10 datasets, considering values ranging from 0.2 to 2 ($\lambda$ equal to 0 is validated in Figure 2 b). The results are shown in the “Ablation study for $\lambda$” and Table 7 in our revised manuscript. Examining the table, we observe that all $\lambda$ values exhibit a similar testing error rate. Notably, $\lambda=1$ consistently delivers the best performance across all methods. Therefore, for consistency, we opt for $\lambda = 1$ in all our experiments.

---

### Official Review · Reviewer_sRgK · 2023-11-01

**Soundness:** 2 fair
**Presentation:** 1 poor
**Contribution:** 2 fair
**Rating:** 5
**Confidence:** 4

**Summary:**

This paper proposes a leader-follower neural network without back-probagation for learning, LFNN, by leveraging local error signals.

LFNN shows better performance than other BP-free variants and achieves <2x speedup compared to BP.

**Strengths:**

The leader-follower neural networks, LFNN, without back-propagation is interesting.

Experiments on CIFAR-10, Tiny-ImageNet and ImageNet are conducted to show the advantages of LFNN.

**Weaknesses:**

The citation reference seems not using ICLR template, e.g., the reference only uses numbers.

The architecture of LFNN is not clear. From the table 1, it looks LFNN took the architecture from [1], LocalMixer.
I would like to know why this is not clearly described in the paper? And why more parameters are added based on [1].

The math definition formula of L_{l}^{\delta} is missing in eq. (1). Also the third term, L_{l}^{\bar{\delta}} is a little bit confusing.
It is a mean squared error. But the number of outputs from leaders seems different from followers in one layer.

In definition 2.1, leaders are decided by prediction errors. But for each hidden layer, how can we get the ground-truth labels?

For the loss, two are using cross-entropy loss and another one is using MSE loss. Are these three losses in a comparable range?

From Table 1, LFNN are outperforming other baselines in [1] on accuracy. How about training time comparing to the baselines in [1]?
Also, what are the insights for the improvements?

In Table 3, why LFNN-l is doing better than BP on Tiny-ImageNet? Is this scalable to ImageNet?


[1] SCALING FORWARD GRADIENT WITH LOCAL LOSSES

**Questions:**

See the weaknesses above.

---

> ### Author Response · Authors · 2023-11-16
> **Response to Reviewer sRgK**
>
> We express gratitude to the reviewer for conducting a thorough assessment of our manuscript and recognizing the merit of our work. In response to each comment, we have provided detailed, point-by-point responses to improve clarity and transparency.
>
> (1) We appreciate the reviewer for raising it out. We have revised our template and updated it to adhere to the correct reference style.
>
> (2) We thank the reviewer for this comment. The models in [1] did not adhere to a classic CNN architecture and we cannot exactly replicate the number of trainable parameters. To ensure a fair comparison, we developed CNN-based architectures using both our LFNN and LFNN-$\ell$ approaches which contain a similar number of trainable parameters as [1].
>
> For a more comprehensive comparison of our LFNN-$\ell$ models with other baseline BP-free models, we integrated both LFNN-$\ell$ and the baseline models into ResNet-101 and -152. This ensured that all models shared the same ResNet architecture as a basis for a fair and consistent comparison (results are presented in Table 3). When applying LFNN-$\ell$ to ResNet, the model will also have additional parameters introduced by the output layers responsible for computing local loss for each block.
>
> [1] Scaling forward gradient with local losses
>
> (3) The term $\mathcal{L}_l^{\delta}$ in our context represents the local prediction loss for leaders, specifically applied between output neurons and leader workers. $\mathcal{L}_l^{\delta}$ is implemented as the cross-entropy loss.
>
> The term $\lambda \mathcal{L}_l^{\bar\delta}$ denotes the local prediction loss for followers, representing the MSE loss between leader and follower workers. In LFNN and LFNN-$\ell$, the followers' loss is not computed between individual leader and follower neurons; instead, it is calculated between leader and follower workers. Each worker comprises the same number of neurons. For instance, referring to Figure 1b, if a layer has 6 neurons and is divided into 3 workers (each worker having 2 neurons), this layer will consist of 1 leader worker and 2 follower workers. The follower loss for each follower worker is then computed with respect to the leader worker. To enhance the clarity of our revised manuscript presentation, we've incorporated the following sentences on pages 2 and 3, respectively: "Unlike classic NNs where neurons are the basic units, LFNN workers serve as basic units."; "A local error signal is applied between leader workers and follower workers." We thank the reviewer for raising these questions to help us improve our manuscript.
>
> (4) We appreciate the reviewer for providing valuable comments that have helped us enhance the clarity of our presentation. In LFNN and LFNN-$\ell$, the hidden layer and output layer use the same ground truth label. To address any confusion related to this aspect, we have revised the "Implementation details" section on Page 4.
>
> (5) We thank the reviewer for asking this question. We have carefully considered the question you raised in your review, but we find that we may need a bit more clarification to fully understand the context or specifics of your inquiry.
>
> If the reviewer meant for us to compare the weight of the loss terms between cross-entropy and MSE in our loss function, we make the hyperparameter $\lambda$ in the function for loss weighting (in equation 2). We add an new section called “Ablation study for $\lambda$” in the revised manuscript where we conducted an ablation study of $\lambda$ for both LFNN and LFNN-$\ell$ on both MNIST and CIFAR-10 datasets, considering values ranging from 0.2 to 2 (lambda equal to 0 is validated in Figure 2 b). Examining the table, we observe that all other $\lambda$ values exhibit a similar testing error rate. Notably, $\lambda=1$ consistently delivers the best performance across all cases. Hence, we choose $\lambda = 1$ in all our experiments. This setting signifies a balanced weight between cross-entropy and MSE loss functions, resulting in optimal performance.

---

> ### Author Response · Authors · 2023-11-16
> **Response to Reviewer sRgK**
>
> (6) We sincerely appreciate the reviewer for bringing up this question. We compare the training time of LFNN-$\ell$ with the best-performing model in [1] (LG-FG-A) on MNIST and CIFAR-10, where both models have an equal number of blocks (set to 4). The results show that LFNN-$\ell$ is 1.56 times faster on MNIST and 1.47 times faster on CIFAR-10 compared to LG-FG-A. These findings not only highlight LFNN-$\ell$'s superior performance over LG-FG-A but also underscore its capability to be trained more expeditiously.
>
> The increased training speed is attributed to the fact that LFNN-$\ell$, in contrast to other block-based BP-free approaches, requires only a small number of additional parameters (as indicated in Table 3c and d). This characteristic minimizes overhead and accelerates the training process.
>
> Moreover, while both LFNN-$\ell$ and LG-FG-A employed self-organizing local connections, it's noteworthy that LG-FG-A depicted interconnections among specific layers and workers (groups of neurons). In contrast, LFNN or LFNN-$\ell$ allows connections among workers, offering a more biologically plausible approach. This connectivity enhancement can facilitate learning with complex input data. Hence, this is the reason that LFNN-$\ell$ can outperform LG-FG-A on a large scale dataset.
>
>
> (7) We appreciate the reviewer for raising this question. As the reviewer correctly noticed, in our experiments, our model exhibits superior performance compared to BP when applied to Tiny-ImageNet using the ResNet architecture. We attribute our model's enhanced performance to the biologically implausible nature of BP. BP does not transmit error signals at a local scale. In contrast, our model is designed to be more biologically plausible by transmitting local error signals (in the human brain, BNNs also transmit signals at local scales). Therefore, given that local transmission in BNNs has been shown to enhance learning with complex input data [2], we posit that this biological plausibility is a contributing factor to the improved performance observed in our model.
>
> Our model is scalable to ImageNet. The experimental results for ImageNet are shown in Table 3 c and d. Tables 3 c and d present the error rates, speedup ratios, and the number of parameters for LFNN-$\ell$ embedded in ResNet-101 and -152 during the analysis of ImageNet, respectively.
>
> [2] Henry Markram, Wulfram Gerstner, and Per Jesper Sjöström. A history of spike-timingdependent plasticity. Frontiers in synaptic neuroscience, 3:4, 2011.

---

### Official Review · Reviewer_KQC9 · 2023-11-06

**Soundness:** 2 fair
**Presentation:** 2 fair
**Contribution:** 2 fair
**Rating:** 5
**Confidence:** 3

**Summary:**

This paper introduces a leader-follower framework that enables neural network learning without the need for backpropagation. Specifically, the authors view deep neural networks as a composition of workers operating at the level of neurons, layers, or blocks. These workers are classified into leaders and followers, with the calculation of individual losses for each. Leader workers use both global and local losses, whereas follower workers are trained to replicate outputs identical to those generated by their corresponding leader workers. The proposed BP-free algorithm abstains from the utilization of global loss for training leader workers. The experiments conducted on MNIST, CIFAR-10, and ImageNet show the potential of the proposed method.

**Strengths:**

1. This paper is overall clearly clarified and well organized.
2. The proposed method offers an interesting approach to enable neural networks to learn without the use of backpropagation.

**Weaknesses:**

1. The proposed method is interesting; however, it lacks a comprehensive explanation of its effectiveness within the neural network learning framework. Specifically, there is a need for detailed elucidation of the mechanism behind the reduction of prediction errors for the leader workers from the intermediate layer, as well as the minimization of the distance between the leader and follower workers.
 - Why does the test accuracy decline between the 40% and 60% leaderships in Figure 2a?
 - In Figure 2b, what role does $L_l^\delta$ serve in the comparison between $L_g$ and $L_g + L_l^\delta$?
 - In Figure 2b, despite the similarity between $L_g + L_l^\delta$ and $L_g + L_l^\delta+L_l^\hat{\delta}$ in the later stages of training, what is the specific function of $L_l^\hat{\delta}$?
2. Insufficient discussion is provided regarding the experimental results.
 - Occasionally, LFNN-$\ell$ leads to superior generalization performance compared to LFNN, as indicated in the results of Table 1. What might be the underlying reason for this phenomenon?
 - While LFNN outperforms BP and LG-BP on simpler datasets in Table 1, it lags behind them in terms of ImageNet results. What could be the reason behind such outcomes?

**Questions:**

(Copied from Weaknesses)
 - Why does the test accuracy decline between the 40% and 60% leaderships in Figure 2a?
 - In Figure 2b, what role does $L_l^\delta$ serve in the comparison between $L_g$ and $L_g + L_l^\delta$?
 - In Figure 2b, despite the similarity between $L_g + L_l^\delta$ and $L_g + L_l^\delta+L_l^\hat{\delta}$ in the later stages of training, what is the specific function of $L_l^\hat{\delta}$?
 - Occasionally, LFNN-$\ell$ leads to superior generalization performance compared to LFNN, as indicated in the results of Table 1. What might be the underlying reason for this phenomenon?
 - While LFNN outperforms BP and LG-BP on simpler datasets in Table 1, it lags behind them in terms of ImageNet results. What could be the reason behind such outcomes?

---

> ### Author Response · Authors · 2023-11-16
> **Response to Reviewer KQC9**
>
> We extend our sincere thanks to the reviewer for conducting a comprehensive assessment of our manuscript. We are genuinely grateful for the acknowledgment of our work as interesting. In addressing each comment, we have provided thorough, point-by-point responses for clarity and transparency.
>
> (1) In Figure 2a, we present the one-pass test accuracy for LFNN on MNIST, representing the output results of the first epoch. It is an interesting observation that test accuracy declines between 40\%-60\% leadership. Given that our LFNN draws inspiration from collective behaviors, and considering that in real-world conditions, only 5\%-10\% of individuals are typically chosen as leaders [1, 2]. We formulate the following potential hypothesis to elucidate this phenomenon: Intuitively, the higher the information leadership size would result in better decision making in complex collectives in nature. However, artificial neural networks are very different from the collective behaviors observed in nature. For challenging tasks, we do observe that higher leadership size tends to give higher accuracy (in Table 5 and 6). In this one-pass training of MNIST, we do not observe this pattern. Our hypothesis is that with few and considerable amounts of leadership, there is less randomness and noisy information during training. With around 50\% leadership, for this small neural network on a simple task with one-pass training, the sub-optimal performance could be caused by the randomness and competing information that followers receive from multiple leaders. We welcome and appreciate any suggestions or comments here.
>
> [1] Peter Friedl, et al. Collective cell migration in morphogenesis and cancer. International Journal of Developmental Biology
>
> [2] Tarcai, Norbert, et al. Patterns, transitions and the role of leaders in the collective dynamics of a simple robotic flock.
>
> (2) Figure 2b displays the results of the ablation study involving various loss functions, namely those using global loss only, global loss with leader loss, global loss with follower loss, and global loss with both leader and follower loss. In this context, $\mathcal{L}_l^{\delta}$ denotes the local prediction loss for leaders, while $\mathcal{L}_g$ represents the global prediction loss. The comparison between $\mathcal{L}_g +\mathcal{L}_l^{\delta}$ and $\mathcal{L}_g$ aims to elucidate whether incorporating local prediction loss for leaders has an impact on the model's performance (refer to "Ablation study of loss terms" on page 5).
>
> (3) The $\mathcal{L}_l^{\bar\delta}$ represents the local error signal of followers which is denoted as $\mathcal{L}_l^{\bar\delta} = \mathcal{D}(a, b)$, where $D$ is the MSE loss function (refer to page 3 for more details).
>
> We would like to express our gratitude to the reviewer for the comment and we add more discussion regarding experimental results in our revised manuscript (in page 9).
>
> (4) As reviewer correctly observed, in Table 1, LFNN-$\ell$ outperformed LFNN exclusively on ImageNet and MNIST, particularly with a leadership size of 70\%. The possible reason underlying this phenomenon is that the local loss structure of LFNN-$\ell$ shares some similarities with BNNs. This structure, characterized by intricate self-organizing local connections, facilitates learning with complex input data [3]. Local loss allows informed leaders in hidden layers to directly expose to the ground truth information and each leader learn collectively similar to the information leadership in the complex collectives in the nature. Hence, we suppose that local loss structure in LFNN-$\ell$ contributes to slightly better performance than BP-based approaches in some cases.
>
> [3] Henry Markram et al. A history of spike-timingdependent plasticity.
>
> (5) It is essential to acknowledge that in the context of BP-free methods, the scalability to larger datasets, presents a notable challenge. In alignment with the previous approaches, although our LFNN also does not outperform BP on larger-scale problems with a small CNN architecture (see Table 1), LFNN-$\ell$ outperforms BP when we increase the complexity of our network. As illustrated in Table 3 (c-d) (applying LFNN-$\ell$ on ResNet), LFNN-$\ell$ shows results that are comparable to those achieved with the BP. Hence, the principal factor contributing to LFNN's relatively inferior performance compared to BP on larger-scale problems appears to be the network size, which is constrained by its relatively small scale. This phenomenon could be attributed to the fact that LFNN and LFNN-$\ell$ require more information provided by leaders when analyzing more complex datasets. In the case of a small CNN architecture, there might not be sufficient information gathered for leaders to effectively guide the training process. Hence, a more complex CNN will include more leaders, which can accumulate enough information to thoroughly analyze the complex data and contribute meaningfully to the training process.

---

> > ### Comment · Reviewer_KQC9 · 2023-11-22
> > **Post-rebuttal**
> >
> > Thanks to the authors for addressing my concerns. However, I still have some concerns regarding the following points:
> > 1. I inquired not about the meaning of each loss but rather about how those losses lead to such results through certain behaviors. However, the authors reiterated the explanations provided in the paper.
> > 2. In response to most of my questions, the authors presented only their hypotheses. However, evidence should follow the presented hypotheses.

---

### Meta-Review · Area_Chair_DzNS · 2023-12-09

**Metareview:**

This paper introduces a framework that enables BP-free training. The idea of making neural network as a composition of followers and leaders with their respective losses is interesting. Experiments on MNIST, CIFAR-10, and ImageNet are conducted to verify the effectiveness. How to get rid of the reliance on back propagation has been an important research problem and attracts wide interests in the community. The method in this paper is technically solid and the performance seems promising. However, the reviewers are concerned about the unclarity of the experimental results. In particular, it is very weird that in Table 1 the BP-free LFNN-l performs much worse than the BP-trained network on ImageNet, but in Table 3 all the BP-free LFNN-l results surpass the BP-trained networks. No evidence is presented to support why the proposed BP-free method LFNN-l can outperform the BP version LFNN. The insufficient discussion of the experimental results makes the study not convincing. The ACs look through the paper and all the comments, and agree with the reviewers. More thorough explanation and transparent experiments are suggested for the future version of this study.

**Justification For Why Not Higher Score:**

More thorough explanation and transparent experiments are needed.

**Justification For Why Not Lower Score:**

N/A

---

### Decision · Program_Chairs · 2024-01-16

Reject